# Voice Cloning: Training Speaker Selection in Limited Multi-Speaker Corpus

*David Guennec, Lily Wadoux, Aghilas Sini, Nelly Barbot, Damien Lolive*

Univ Rennes, CNRS, IRISA, France

{`firstname.secondname`}@irisa.fr

## Abstract

Text-To-Speech synthesis with few data is a challenging task, in particular when choosing the target speaker is not an option. Voice cloning is a popular method to alleviate these issues using only a few minutes of target speech. To do this, the model must first be trained on a large corpus of thousands of hours and hundreds of speakers. In this paper, we tackle the challenge of cloning voices with a much smaller corpus, using both the speaker adaptation and speaker encoding methods. We study the impact of selecting our training speakers based on their similarity to the targets. We train models using only the training speakers closest/farthest to our targets in terms of speaker similarity from a pool of 14 speakers. We show that the selection of speakers in the training set has an impact on the similarity to the target speaker. The effect is more prominent for speaker encoding than adaptation. However, it remains nuanced when it comes to naturalness.

**Index Terms**: Voice Cloning, Speaker Adaptation, Speaker Encoding, Speaker Similarity

## 1. Introduction

Modern Text-To-Speech (TTS) systems are mostly neural-based approaches and share a common framework, a neural acoustic model [1, 2, 3], for inferring an intermediate representation of the speech signal, usually a mel-spectrogram, which is then provided to a neural vocoder [4, 5, 6] for predicting the speech signal.

Traditional neural TTS builds an acoustic model for a single voice. This approach is costly as it requires a substantial quantity of records from the target voice. However, in certain scenarios, the data per speaker is limited and/or publicly unavailable, which pushes the TTS community to find alternatives. Multi-speaker TTS [7, 8] is one possible alternative as it is capable of generating speech of a voice seen during the training process. Other Voice Cloning techniques, such as speaker adaptation methods [9, 10, 11] or speaker encoding methods [12], are capable of generating speech from only a few samples from a new speaker.

Speaker adaptation [9] is the process of fine-tuning a pre-trained multi-speaker acoustic model, in order to specialize it so that it only produces the target speaker's voice. Each new speaker therefore requires a fine-tuning step on the pre-trained model in order to obtain a custom model. This method may only affect speaker-related parameters, or the complete model.

Conversely, the speaker encoding method [12] is trained only on a multi-speaker corpus, and does not require any fine-tuning. Instead, a second model, called speaker encoder, provides the acoustic model with a vector representation of speaker features, called speaker embedding [13, 14]. To match another speaker, new audio samples are provided as input to the speaker encoder.

Both approaches require relatively few speech samples from the target speaker, resulting in very good results with ten minutes of speech and good results with ten seconds [11]. While speaker encoding has slightly lower results compared to speaker adaptation [10], both appear promising. When applicable, these methods can also be combined : a speaker encoding model can be fine-tuned to a target speaker.

Usually, multi-speaker acoustic models are trained on proprietary or well-established multi-speaker corpora, such as VCTK [15] or LibriTTS [16]. Data is controlled, of high acoustic quality and engineered specifically for speech applications. It contains hundreds to thousands of both hours of speech and speakers. To train a voice cloning system in a language other than English though, the choice of the dataset can be more difficult. In French, multi-speaker datasets dedicated to TTS are not easily available, let alone with enough speakers and quality for voice cloning. Well-established corpora exist, such as the FrenchSiwis [17], FHarvard [18] and Bref corpora [19] but they often contain only one or few speakers, can be of various quality, not adapted for multi-speaker speech synthesis, or proprietary. In this context of relative scarcity of data, finding a high number of quality speakers for training is difficult. When the training corpus is composed of hundred to thousand of speakers, it seems logical to train the model on all available speakers so that it learns better to generalize to unknown speakers. Nevertheless, when the dataset is limited, training on all samples available, including speakers very different from the target, could make the task harder for the model.

In this study, we consider the case where only a few samples are available for the target speaker. We further assume that resources are limited to train the acoustic model, i.e. large and high-quality multi-speaker corpora are not readily available. Thus, training speaker specific TTS model from scratch is not possible. In this context, voice cloning appears to be a viable option for reproducing the voice of the target speaker. Consequently, this work aims to investigate the performance of both voice cloning approaches, speaker adaptation, and speaker encoding under a constrained multi-speaker training corpus with few speakers available. In particular, we investigate whether the similarity of the training corpus to the target speaker impacts the performance of the TTS system. To achieve this, we evaluate the impact of speaker selection on two objective evaluation metrics, such as speech quality and speaker similarity. A DMOS subjective test is also conducted to assess speaker similarity. It is important to note that selecting data based on a target speaker is not always ideal, as the process might impede the model if the target voice changes. In our case, this is not an issue as our aim is to assess the impact of selecting data based

on the target.

The rest of the paper is organized as follows. In Section 2, the two voice cloning methods are introduced before presenting the datasets and the proposed selection method in Section 3. The training procedure and details are then provided in Section 4. Finally results are discussed in Section 5.

# 2. Voice Cloning Systems

The goal of voice cloning systems is to reproduce the voice of a known target speaker. Here, we introduce the two methods under assessment in the paper, namely speaker adaptation and speaker encoding.

## 2.1. Speaker Adaptation

The first method, speaker adaptation [10, 11], requires a pre-existing multispeaker acoustic model. Audio-text pairs from a never seen speaker is then used to fine-tune the acoustic model to that speaker. These speaker specific audio-text pairs are generally no more than a few minutes when combined and may feature varying audio quality. Fine-tuning is only run for a few epochs to avoid overfitting. It can be done on the entire model or only on a portion of its layers or components while others are frozen. In this paper, we fine-tune the entire acoustic model. The multispeaker vocoder, which produces the final audio samples from the output of the acoustic model, is left unchanged.

In this paper, we train a Tacotron2 acoustic model [1] with multispeaker audio using the popular toolkit ESPNet [20]. Training is done based on the *LJSpeech* recipe as no speaker encoder is used for speaker adaptation. The Tacotron2 acoustic model transforms textual input into mel-spectrograms, and classically follows an encoder-decoder architecture. We use phonetic inputs, computed from text with the eSpeak toolkit[1].

The same vocoder model is used for speaker adaptation and speaker encoding (presented in the next section). For both approaches, WaveGlow [4] is thus used instead of the original WaveNet of Tacotron2, for faster inference. The official NVidia implementation[2] is used.

## 2.2. Speaker Encoding

For the speaker encoding method [12], the considered approach, displayed in figure 1, relies on two models, a speaker encoder model transmitting a speaker embedding, assumed to contain vocal identity, to a multi-speaker TTS model, composed of an acoustic model and a vocoder.

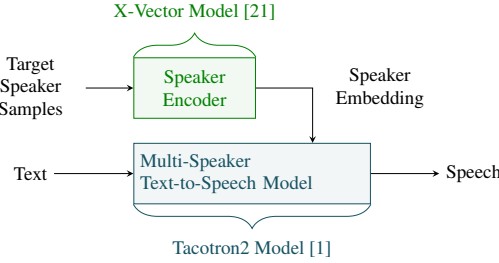

Figure 1: *Speaker Encoding Voice Cloning System.*

In voice cloning, speaker encoders usually derive from speaker classification or speaker verification tasks. Speaker classification aims at determining which speaker a speech sample originates from, within a fixed set of speakers. Speaker verification seeks to determine whether two given speech samples are from the same speaker. The x-vector model [21] is frequently used for this task. Extracted speaker embeddings, called x-vectors, correspond to a segment-level layer embedding. The implementation used in this article is from Kaldi[3]. For a condensed representation, the chosen embedding dimension is 32, and the model layer dimensions are scaled down accordingly.

The model used in this article is a multi-speaker version of Tacotron2 [1], in which the speaker embedding is concatenated to the output of the encoder, then transmitted as input to the decoder. The same phonetic inputs are used as for speaker adaptation. In this paper, we do not perform a fine-tuning as would be done for speaker adaptation in order to be able to compare the respective merits of each approach in our results but such a combination is worth considering in a future analysis.

# 3. Datasets

In this section, we present the different corpora used for the study. In particular, after having described the source of our training dataset, test speakers are detailed. Finally, we describe the way training corpora are composed to evaluate the impact of similar voice on the quality and similarity criteria.

## 3.1. Mufasa Corpus

All training data used in this study is extracted from the Mufasa corpus. The Mufasa corpus[4] is an evolutive database of audiobooks' text and audio samples, from public domain audiobooks in French language, recorded by private individuals. The version used in this study is composed of 14 speakers - seven male and seven female speakers.

We use a perception-related automatic measure as a criterion to filter the corpus, in order to keep only good quality samples. MOSNet [22] is a recent neural-based automatic evaluation metric, trained to predict MOS perceptual scores, originally developed for voice conversion tasks. In this study, we prefer to use the WV-MOS model [23], which uses a wav2vec2.0 model [24] instead of the standard MOSNet architecture, to improve prediction proficiency. It is trained on the same data as standard MOSNet. The trained model used is available with the implementation[5]. We found that a 3.5 threshold offers, for this corpus, a good trade-off between acoustic quality and available quantity per speaker Therefore, all samples with a WV-MOS score lower than 3.5 are discarded.

To ensure equivalent representation for each speaker in the training set, only 1 hour of randomly selected samples is kept per speaker. This choice of 1 hour was made for two reasons. First, to prevent having to reduce further the number of available speakers due to a lack of data (some speaker have little more than 1h left after filtering out natural samples with a mos score lower than 3.5). Second, to favor the constitution of datasets with a number of individual speakers as large as possible in order to mitigate the potential impact of individual speakers on our results.

---

[1]http://espeak.sourceforge.net
[2]https://github.com/NVIDIA/waveglow

[3]https://github.com/kaldi-asr/kaldi/tree/master/egs/sre16/v2
[4]https://sites.google.com/view/mufasa-corpus
[5]https://github.com/AndreevP/wvmos

### 3.2. Test Speakers

In our experiments, we use three test speakers coming from various corpora and with different properties:

- KG: female speaker, was recorded for Text-to-Speech, with controlled speech and expressiveness. The samples are high-quality and manually verified.
- FHarvardM: male speaker from the FHarvard corpus [25] containing phonemically balanced speech samples. Recordings are intended for speech applications. His voice is calm and steady.
- Cocotte: female speaker, originally from Mufasa, but not included in our training speakers as too few of its samples had a WV-MOS score > 3.5. This is a senior speaker, tranquil, with a southern french accent and a slight lisp.

For each one of these speakers, we extracted two datasets: a 10 min reference set for the speaker, used both for fine-tuning and as input to the speaker encoder and a test set, containing text to synthesize and its corresponding natural audio. These sets were constructed by randomly selecting samples with a WV-MOS score greater than 3.5, corresponding to utterances with an average duration of 3.72s, until the total duration of samples selected for this speaker reaches 10 minutes. For speaker adaptation, these samples are used to fine-tune the trained model to the target speaker. Their corresponding phonetised transcriptions, provided by espeak, are also used. For speaker encoding, these samples are first concatenated to obtain a unique sample and then given as input to the speaker encoder in order to extract the speaker embedding.

In the case of KG and FHarvardM, test sets are constituted of 100 samples and their transcription, selected randomly from the data unused for reference sets. As not enough data was available for Cocotte, we created a 30 samples set using all remaining data available.

### 3.3. Training Corpora

After filtering with a 3.5 WV-MOS score threshold, speaker similarity between each one of the test speakers and the 14 potential training speakers from Mufasa was evaluated automatically with the Resemblyzer model [14]. Resemblyzer is a speaker encoder producing 256-dimensions embeddings. The model used was the pre-trained model available in the implementation repository[6]. A 10-minute reference sample is randomly built by concatenation for each speaker. For the test speakers, this sample corresponds to the reference sample later used as speaker encoder input. From each of these samples, a resemblyzer embedding is extracted and cosine similarity is computed two-by-two between these embeddings.

Using these measures we created 4 subsets of our selection of Mufasa for each of our 3 target speakers. The first subset is comprised of the *5 closest training speakers* to the target speaker irrespective of gender. This represents the training material that is closest to the related training speaker. The second subset is similar, but only considers speakers of the same gender as the associated target speaker. This subset of the *5 closest speakers of the same gender as the target* is meant to help us check the impact of gender on similarity, when possible. Indeed, for target speakers KG and FHarvardM, both subsets are identical as similarity to training speakers is largely correlated to gender for these two targets. In consequence, these sets are one and the same for KG and FHarvardM. However, this is not

---

[6]https://github.com/resemble-ai/Resemblyzer

the case for target speaker Cocotte, which does have two distinct subsets. In fact, as displayed in table 1, the 5 closest speakers subset has more speakers of the opposite gender than of the same (3 male, 2 female). The two remaining subsets we consider in this study are meant to investigate what happens when training data is dissimilar to the target. These sets are the *5 farthest training speakers irrespective of the gender* and the *5 farthest training speakers of the same gender as the target*. Again, having the two subsets is meant to verify the impact of gender on final similarity in generated speech.

Thus, in total, we have 3 datasets for KG (closest, farthest & farthest same gender), 3 for FHarvardM (closest, farthest & farthest same gender) and 4 for Cocotte (closest, closest same gender, farthest & farthest same gender). The training speakers composing each set is shown on table 1. In addition to these 10 training sets, we created a $11^{th}$ by aggregating all 14 training speakers together. This last training set is called "All speakers" in the table. As it contains all data from all training speakers, this last set also has a different duration overall: 14h versus 5h for all others (as every training speaker has 1h).

It is important to note that given the modest size of the dataset used in this study, some of the subsets we formed have some speakers in common, even when said subsets have opposite semantics. For instance, given that target speaker FHarvardM has better similarity scores with male speakers and that we have 7 male speakers in total available for training, the 5 closest and 5 farthest datasets of the same gender do have 3 speakers in common. Overall, the presence of such mixed sets is an opportunity to see if only partial changes in the composition of a corpus can lead to substantial gains/losses in cloned audio naturalness and speaker similarity. It must be noted though that datasets labeled as closest and farthest irrespective of gender never have a speaker in common. This only appears when including same gender datasets in the comparison.

Finally, our 11 datasets are used to train:

- 13 speaker adaptation models (as the "All speakers" dataset is fine-tuned for each one of the 3 target speakers).
- 11 speaker encoding models (one for each dataset). Each model is then provided with the relevant speaker embedding.

In both case, this leads to a total of 13 separate evaluations as represented in tables 1 and 2.

## 4. Training

Speaker adaptation and speaker encoding approaches are detailed in this section thus providing details to favour reproducibility. At the end, a details concerning the vocoder are given. One can note that the vocoder is common to both approaches.

### 4.1. Speaker Adaptation Models

As described in section 2.1, the acoustic model used for both speaker adaptation and encoding is the Tacotron2 architecture. For speaker adaptation, the model is trained with multispeaker data but has no speaker encoder as it does not need one.

The Tacotron2 acoustic model, which is more sensitive to corpus quality than the vocoder or speaker encoder (see section 4.2), is trained in two phases. First, as in [26], a pre-training step is applied with a clean mono-speaker corpus to give the acoustic model a "warm start". The corpus used is FrenchSiwis [17], containing high-quality audio samples from a French female speaker and their transcriptions, aimed at speech synthesis.

| Target speaker | | Model | Gender | Selected speakers | WV-MOS | Avg. cosine similarity |
|---|---|---|---|---|---|---|
| Name | WV-MOS | | | | | |
| *KG* | 4.20 ±0.04 | Closest | All (S) | *Julie*, *Naf*, *Cecile*, *Corinne*, *Pomme* | 4.04 ± 0.01 | 0.82 |
| | | Farthest | Same | *Cecile*, *Corinne*, *Pomme*, *Ezwa*, *Orangeno* | 3.96 ± 0.01 | 0.78 |
| | | Farthest | All | Graigolin, Damien, Bernard, Didier, Daniel | 3.84 ± 0.01 | 0.62 |
| | | All speakers | All | All 14 available training speakers | 3.96 ± 0.01 | 0.72 |
| FHarvardM | 4.33 ±0.03 | Closest | All (S) | Bernard, Daniel, Graigolin, Alain, Dousset | 3.97 ± 0.01 | 0.76 |
| | | Farthest | Same | Graigolin, Alain, Dousset, Didier, Damien | 4.02 ± 0.01 | 0.70 |
| | | Farthest | All | *Naf*, *Pomme*, *Orangeno*, *Ezwa*, *Corinne* | 3.94 ± 0.01 | 0.55 |
| | | All speakers | All | All 14 available training speakers | 3.96 ± 0.01 | 0.65 |
| *Cocotte* | 3.65 ±0.02 | Closest | All | *Pomme*, Damien, *Ezwa*, Dousset, Alain | 3.93 ± 0.01 | 0.74 |
| | | Closest | Same | *Pomme*, *Ezwa*, *Cecile*, *Orangeno*, *Naf* | 3.90 ± 0.01 | 0.70 |
| | | Farthest | Same | *Cecile*, *Orangeno*, *Naf*, *Julie*, *Corinne* | 4.06 ± 0.01 | 0.64 |
| | | Farthest | All | *Naf*, *Julie*, Daniel, Bernard, *Corinne* | 3.96 ± 0.01 | 0.61 |
| | | All speakers | All | All 14 available training speakers | 3.96 ± 0.01 | 0.67 |

Table 1: *Selected speakers, WV-MOS scores and average similarity to the target speaker for each training set. The second column gives the average WV-MOS (with confidence intervals at 95% provided by a bootstrap method) of samples that compose the reference set. Selected speakers are ranked in order of decreasing similarity to the target. It is important to note that, for a same target speaker, some training sets have speakers in common (underlined in the table). The mention "(S)" in column gender means that selected training speakers are the same whether or not we restrict the selection to same-gender speakers. Names in italic text correspond to female speakers and non-italic to males.*

All speaker adaptation models were trained on a NVIDIA GeForce RTX 3080 Ti GPU. A first model trained for 60 epochs on FrenchSiwis (patience mechanism set to 10) with a learning rate set to $1e^{-3}$. The subsequent training of the models on our 11 sets used a learning rate of $5e^{-4}$ and lasted for an additional 20 to 34 epochs depending on the model. 11 models were trained during that step, one for each training set described in section 3.3.

Finally, 13 separate fine-tunings were performed on the models using the 10 minutes data from our target speakers described in section 3.2. 4 fine-tuning operations were done for KG (closest, farthest, farthest same gender and all speakers), the same for FHarvardM and Cocotte with an additional one for this last target (closest same gender). Again, table 1 provides the combination of training sets and target speakers used. The learning rate was set to $1e^{-4}$.

### 4.2. Speaker Encoding Models

For speaker encoding, as the data requirements are different between the acoustic model and the speaker encoder, the two are trained in different steps as in [12]. As the evaluation deals with the impact of the acoustic model training corpus, the speaker encoder, trained separately, is common to all model versions. Likewise, the acoustic model and vocoder are trained separately (as the same vocoder model is used for all systems in this paper). All models are trained on a NVIDIA V100 GPU.

Speaker encoder training requires a high number of speakers. Yet in [12], it seems more resistant to signal noise than the acoustic model. Thus, it can be trained with lesser quality signals. We assume their conclusions to be extendable to other speaker verification encoders such as the x-vector model. This hypothesis serves as a basis to choose its training corpus. The x-vector model used here is trained from scratch on the French version of the CommonVoice corpus [27]. CommonVoice is an open-source multi-lingual corpus by Mozilla, allowing volunteers to record speech samples via their recording device. The version used contains 1007 hours of validated data

and 16785 speakers, which is consistent with corpora used in state of the art voice cloning. Transcriptions belong to a pool of more than 2 million sentences, ensuring sample content diversity. Nevertheless, transcriptions are not given to the model, as it is text-independent. Due to the diversity of recording devices and background sound environments, sample quality is very variable. For reproducibility, the default train, dev and test sets are kept.

As for speaker adaptation, a two phases training is performed, starting with a pretraining on the FrenchSiwis corpus. The Tacotron2 model trained for 48 epochs, and stops with a patience mechanism set to 10 epochs. From this pre-trained model, one model is trained on each of the subsets presented in section 3. They stop with a patience of 10, between epochs 71 and 79, including the 48 epochs of pre-training.

### 4.3. Vocoder

All models presented in this section use the same vocoder, presented in section 2. Vocoder training is a longer, costly step which may a week or more. Thus, the WaveGlow model used in the experiments is not trained from scratch, but fine-tuned on French from the official English model. This second training phase is executed on the complete Mufasa corpus, presented in section 3, without WV-MOS score selection. The model was trained for 40 epochs (372500 batches) using default hyperparameters.

## 5. Results

Our main hypothesis is that favouring, in the training corpus, the similarity to the target speaker rather than the number and the variety of the speakers, for the same volume of data per speaker and a comparable quality, improves the similarity of the cloned samples without degrading naturalness. The relative importance of speaker similarity and audio quality is thus our main point of study. However, a potential bias might come from the fact that some speakers may be more suited for TTS than others. We be-

| Target speaker | Training speakers | Gender | WV-MOS | | Cosine Similarity | |
|---|---|---|---|---|---|---|
| | | | Adaptation | Encoding | Adaptation | Encoding |
| *KG* | Closest | All (S) | $3.47 \pm 0.10$ | $3.25 \pm 0.08$ | $\mathbf{0.750 \pm 0.015}$ | $\mathbf{0.664 \pm 0.014}$ |
| | Farthest | Same | $\mathbf{3.65 \pm 0.09}$ | $3.36 \pm 0.08$ | $\mathbf{0.755 \pm 0.012}$ | $0.632 \pm 0.011$ |
| | Farthest | All | $3.46 \pm 0.09$ | $3.33 \pm 0.13$ | $0.741 \pm 0.014$ | $0.512 \pm 0.011$ |
| | All speakers | All | $\mathbf{3.67 \pm 0.08}$ | $\mathbf{3.51 \pm 0.07}$ | $0.739 \pm 0.015$ | $\mathbf{0.675 \pm 0.013}$ |
| FHarvardM | Closest | All (S) | $\mathbf{3.92 \pm 0.04}$ | $3.10 \pm 0.05$ | $\mathbf{0.892 \pm 0.005}$ | $\mathbf{0.688 \pm 0.008}$ |
| | Farthest | Same | $3.87 \pm 0.04$ | $\mathbf{3.61 \pm 0.07}$ | $\mathbf{0.898 \pm 0.004}$ | $0.591 \pm 0.008$ |
| | Farthest | All | $3.67 \pm 0.04$ | $3.39 \pm 0.06$ | $0.874 \pm 0.005$ | $0.437 \pm 0.008$ |
| | All speakers | All | $\mathbf{3.91 \pm 0.04}$ | $3.10 \pm 0.06$ | $0.887 \pm 0.005$ | $0.662 \pm 0.010$ |
| *Cocotte* | Closest | All | $\mathbf{2.89 \pm 0.10}$ | $3.13 \pm 0.13$ | $\mathbf{0.784 \pm 0.020}$ | $\mathbf{0.623 \pm 0.019}$ |
| | Closest | Same | $2.69 \pm 0.18$ | $3.28 \pm 0.11$ | $0.753 \pm 0.020$ | $\mathbf{0.609 \pm 0.024}$ |
| | Farthest | Same | $2.49 \pm 0.25$ | $\mathbf{3.68 \pm 0.13}$ | $0.760 \pm 0.021$ | $0.508 \pm 0.017$ |
| | Farthest | All | $2.66 \pm 0.15$ | $3.27 \pm 0.13$ | $0.761 \pm 0.020$ | $0.535 \pm 0.025$ |
| | All speakers | All | $\mathbf{2.93 \pm 0.16}$ | $3.30 \pm 0.15$ | $0.756 \pm 0.026$ | $0.585 \pm 0.023$ |

Table 2: *WV-MOS scores and cosine similarity between Resemblyzer embeddings for Speaker Adaptation Models and Speaker Encoding ones with confidence intervals at 95% (provided by a bootstrap method). Scores in bold correspond to the best results per column and target speakers. Several results are highlighted when results are particularly close.*

lieve the WV-MOS filtering above 3.5 on training data should help mitigate this concern. Here, we evaluate audio generated by speaker adaptation and encoding models for all 13 use cases presented at the end of section 3.3, both in terms of similarity to the original speaker and naturalness (WV-MOS). As we did for training data, cosine similarity obtained with Resemblyzer embeddings is used to assess speaker similarity and WV-MOS is used for naturalness. In addition to these objective measures, a DMOS subjective test is performed to further evaluate speaker similarity. As naturalness is not the main subject of this paper and as WV-MOS already serves the same purpose as a subjective evaluation of speech quality and results in the literature, as well as our own experience, suggest that it compares well with evaluations performed by human subjects, we elected not to conduct a subjective test for naturalness.

## 5.1. Speaker Similarity Evaluation

### 5.1.1. Objective Evaluation

Cosine similarity scores obtained with the speaker adaptation approach are given in table 2 in the corresponding column on the left hand side. Correspondingly, results for speaker encoding are on the right hand side of the same column.

Overall, results show that similarity of the training speakers to the targets does indeed have a beneficial or non negative impact on cloned voices with both methods. This is especially helpful for speaker encoding as results for this method are globally much lower than speaker adaptation. It is thus possible that the impact of similarity for adaptation is less important due to these already high scores.

For all targets and speaker adaptation systems, cosine similarity to target speakers is relatively high from 0.739 to 0.898. For targets KG and FHarvardM, we observe that the closest and the farthest speakers (same gender) come on top. For these target speakers, 60% of the content of the closest and the farthest speakers corpora is common (see table 1). However, as, for KG and FHarvardM, similarity scores seem to be closely tied to gender, this might be an indication that the superior score is mainly driven by same gender speakers and that smaller, intra-gender variability has a lesser impact. Speaker Cocotte is different though, since gender and speaker similarity are not closely

related. In that case, the closest voices prove superior to all other models.

For speaker encoding, scores are significantly lower than adaptation ranging from 0.508 to 0.688 across systems. The closest speakers obtain the highest scores in two cases (FHarvardM and Cocotte) and the second highest in the third case (KG). However, unlike the results we obtained for speaker adaptation, the farthest speakers of the gender consistently perform worse than both the closest and the "all speakers" models. This is an indication that similarity indeed helps considerably in the case of speaker encoding. The fact that the "all speakers" models also consistently outperforms the farthest speakers indicates that the model was not overly hindered by the training on dissimilar speakers.

### 5.1.2. Perceptual Evaluation

In order to validate the objective results obtained with cosine similarity, we also conducted a DMOS perceptual evaluation. This evaluation was done for all target speakers but models trained on same genre data were not included in order to reduce the number of samples to evaluate. As a result, 6 systems are evaluated for each target speaker : Closest (All genders), Farthest (All genders) and All speakers, first using speaker adaptation and then speaker encoding. The evaluation was completed by 12 fluent speakers of French (most were native speakers) using the web-based FlexEval[7] evaluation platform [28]. All testers used headphones.

Results are displayed in fig. 2. For each approach, systems trained with data from "closest" systematically achieve a better score than those trained on the "farthest" data, although in some instances a confidence interval overlap occurs. This trend is particularly visible when it comes to speaker encoding (especially in the case of KG and FHarvardM). This is largely similar to observations made with cosine similarity. Likewise, we also observe in the DMOS that differences between "closest" and "farthest" are less significant when considering results for speaker adaptation. Models trained from data from all speakers tend to obtain results close to those obtained by the models

---
[7]https://gitlab.inria.fr/expression/tools/FlexEval

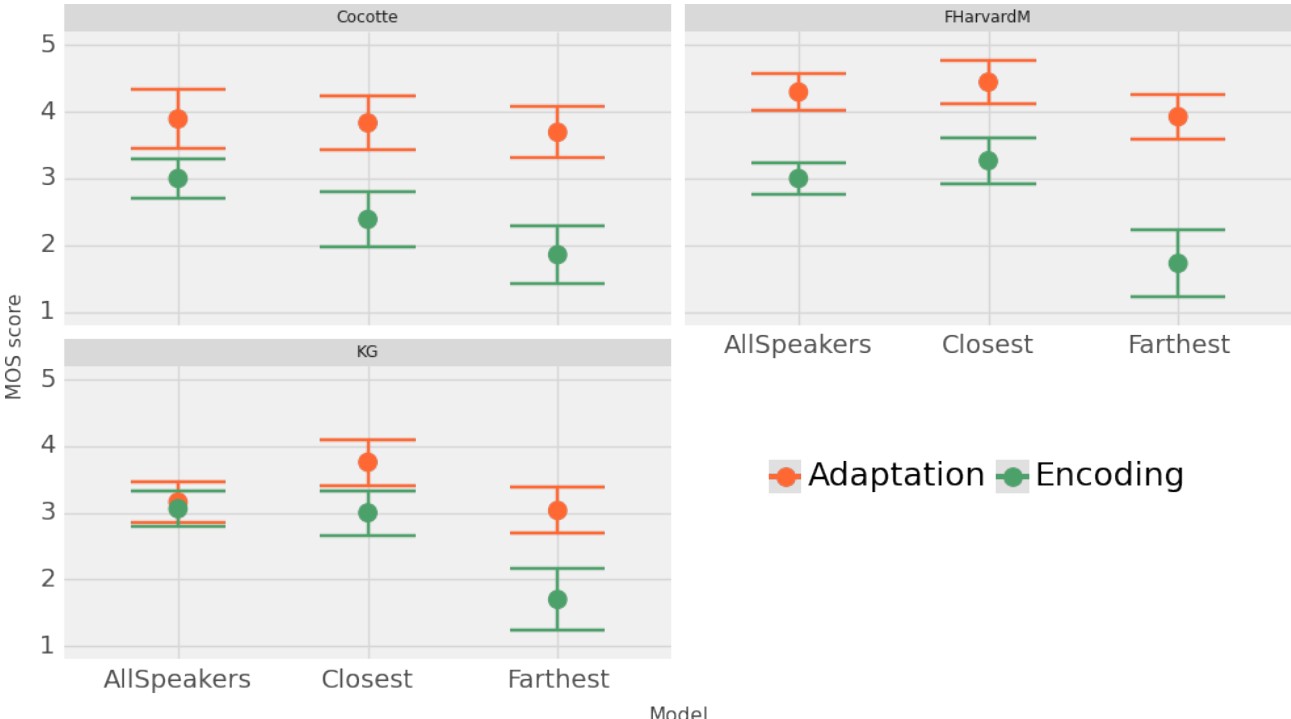

Figure 2: *Results of the DMOS perceptual test. Results for Cocotte are in the top-left corner, FHarvardM in the top-right and KG in the bottom-left corner. Confidence intervals at 95% are computed with a bootstrap method.*

trained on the closest data.

Speaker encoding results for Cocotte for instance show that the "all speakers" model fares better than the "closest" one, which is the opposite of what was observed for cosine similarity. Overall, results for target speaker Cocotte are more nuanced than what was observed with cosine similarity. It is possible that, as the speaker is atypical, the model from "all speakers" learned a superior representation of the speaker space thanks to its supplement of data which enabled it to outperform the model trained on "closest".

When it comes to FHarvardM and especially KG though, this is the opposite: results seem more pronounced that what we saw with cosine similarity. For KG, when using speaker adaptation, "closest" for instance, fares much better than "All speakers".

Overall, these results tend to confirm our conclusions : training only on data from the closest speakers yields similar and sometimes even better results than training on "All data", even though "All data" has nearly 3 times more material.

### 5.2. Quality Evaluation

Results for the naturalness evaluation are presented in column "WV-MOS" in table 2.

As for speaker adaptation, data quantity does not seem to provide a substantial difference as, for each test speaker, there is at least one other system that performs as well as our "all speakers" model. In particular, we observe that both the models trained on speakers closest and farthest (all genders) to KG perform significantly worse than with the farthest (same gender) and all speakers. This can be astonishing especially as training data for closest and farthest (same gender) favor the former (see WV-MOS score in table 1). What is even more interesting

is that only two speakers out of 5 differentiate these sets (Julie and Naf vs. Orangeno and Ezwa). We believe that reading style is the main reason for the difference here. Indeed, speaker Ezwa has a relatively neutral and very regular style while speaker Naf features much more variability thus making it less suitable for acoustic model training than Ezwa. This hypothesis is reinforced by observations on speaker Cocotte where all voices including Naf seem to fare worse than those including Ezwa.

On the contrary, results for FHarvardM show that the model trained on farthest (all genders) is clearly worse than the other configurations. Here, it would seem that gender does provide an improvement. When it comes to Cocotte, results are more difficult to interpret due to the large confidence intervals (as we only have 30 test sentences for this speaker) and lower naturalness of synthesized samples. From the tendencies we observe though, the model trained with the closest speakers seems to perform as well as "all speakers" and better than all others. In that instance, gender does not appear to play a role.

Results for speech encoding follow a similar trend for KG although the "all speakers" system performs much better. For the two other speakers however, results are quite different. For both FHarvardM and Cocotte, the "Farthest (same gender)" achieves a much better score than others and "all speakers" does not appear to perform particularly well. It could be explained by the better average WV-MOS score of the "Farthest (same gender)" training corpora. Overall, speaker similarity does not reflect better naturalness. On the contrary, models trained on the closest speakers fare worse than other systems.

Finally, adaptation appears to be much more sensitive to the naturalness of target speech when compared to speaker encoding. In this respect, natural samples for Cocotte, which have a MOS score of 3.65, lead to a much lower MOS score when using adaptation than what is obtained for KG and FHar-

vardM (MOS of 4.20 and 4.33 respectively).

To sum up, selecting the training dataset based on speaker similarity to the targets leads to contrasted results for speaker adaptation. Speaker selection seems to help but data quantity and speaker style also seem impactful. More experiments are needed to confirm this. However, when it comes to speaker encoding, results suggest that speaker similarity does not help to improve the naturalness of cloned samples. Adding training data and/or speakers does not seem to be particularly effective in our results.

## 6. Conclusion

In this paper, we evaluated the impact of the choice of training speakers for voice cloning by both speaker adaptation and speaker encoding. We showed that while this does not seem to have a clear impact on naturalness, it is helpful when it comes to improving the similarity to the target speaker, especially for speaker encoding. Similarity specifically seems the more important factor when opposed to just gender. We also highlighted that more data is not always helpful. Furthermore, we observed that speaker adaptation is quite sensitive to the naturalness of target samples, an effect that does not occur for encoding.

As this study is (intentionally) conducted on a relatively small training corpus, we aim at expanding our analysis with new speakers and include the speaking style in order to both quantify its impact and evaluate more closely its importance relative to similarity, gender and data quantity.

## 7. Acknowledgements

The authors would like to thank Antoine Perquin for his advice and help in proofing this paper. We thank the reviewers for their insightful observations and advice. This work was granted access to the HPC resources of IDRIS under the allocation 2023-AD011011870R2 made by GENCI.

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
