# OpenReview forum: "Voice Cloning: Training Speaker Selection with Limited Multi-Speaker Corpus"
_Interspeech.org/2023/Workshop/SSW — SSW12_

### Official Review · Reviewer_9tZD · 2023-06-04
**Interesting experiment but not really conclusive**

**Rating:** 4
**Confidence:** 4

**Review:**

The authors consider the scenario of voice cloning when there is not a large dataset of speakers to train the initial multispeaker model, and investigate whether it is useful or not to select the training speakers based on their similarity to the target. It's an interesting question although it might be argued that selecting the training speakers based on the target is not really practical, as one does not want to retrain a different multispeaker model for each target voice to be cloned. Also the distinction between speaker adaptation and speaker encoding methods is a bit artificial, as both methods can perfectly be combined (ie using a multispeaker TTS model with pre-trained speaker embeddings and doing fine-tuning on top of it). Finally the authors don't mention that there could be several ways to do fine-tuning (ie by freezing some layers and only fine tuning the decoder layers).

One interesting aspect of the paper is the use of perception-related automatic measures to filter out the training dataset using the WV-MOS model. The experiment is well presented but it lacks a true subjective test. The authors use two automatic metrics instead, the WV-MOS-based naturalness score and a similarity score (cosine distance between Resemblyzer embeddings). We feel that the lack of true subjective tests is detrimental here.

Finally the experimental results do not really support the initial hypothesis that it is better to choose training speakers close to the target.
The results vary a lot depending on the test corpus, but on the KG corpus, the best similarity score for the speaker encoding approach is obtained on the model trained on all speakers. One explanation (not given in the paper) could be that training on all speakers help the multispeaker model to learn a better representation (more interpolable) of the speaker space.

---

### Official Review · Reviewer_Yi4B · 2023-06-09
**Work of limited contributions, sufficient for publication**

**Rating:** 6
**Confidence:** 4

**Review:**

The authors present a work in the applicative field of voice cloning aiming to evaluate the effect of limited training resources in terms of both target speaker similarity and naturalness.

In particular they work with speech in French language, arguing the limited/restricted resources available to train multi-speaker systems in this language.

Two approaches are used for voice cloning: speaker adaptation and speaker-encoding based frameworks.

The authors use audiobook content for training. 14 speakers, one hour each. Three different speakers are used for testing.

Automatic quality evaluation metrics (WV-MOS model) are used to filter low quality data.

A variety of training subsets (evaluation cases) were defined based on the similarity (closer, farthest) and gender matching between training and testing speakers. Speakers overlap existed between training subsets due to the designed training and evaluation conditions.

The automatic WV-MOS and Cosine similarity metrics were evaluated among the designated trainig subsets / testing speakers. Although the limited amount of data and speakers does not allow to highlight strong conclusions but only particular case-dependent trends, the limited scope of the experimental evaluation still provides some information, strategy, and results, that may be useful in the understanding of the voice cloning problem.

---

### Official Review · Reviewer_tW1g · 2023-06-13
**Experiments are reasonable, but technical novelty is limited**

**Rating:** 6
**Confidence:** 4

**Review:**

The aim of this paper is to investigate the performance of voice cloning approaches for text-to-speech (TTS) synthesis, i.e., speaker adaptation and speaker encoding, when only a small number of samples are available for the target speaker and the scale of multi-speaker corpora is limited. Specifically, to investigate how much the choice of speakers used for training affects the performance of the synthesized speech, comparative experiments are conducted using the Mufasa corpus as the training dataset and Tacotron2 as the base TTS system. The experiments are well designed and the conclusions drawn from the results are reasonable and insightful. However, while I am aware that it is not the purpose of this paper to propose new technical ideas, the technical novelty is limited.

---

### Author Response · Authors · 2023-07-12
**Response to reviewers**

The authors wish to thank the reviewers for their insightful observations and advice. We do agree with the reviewers' critics and have worked on the paper to address as much of it as possible, as detailed in the remainder of this response.

We agree that results presented in this paper should ultimately be supported by more extensive experimentation as uncertainties remain due to the small size of the datasets we use. A new experiment was added to the paper and a more ambitious study, using a larger body of training speakers to choose from, is currently ongoing for a follow up paper.

We amended the paper to address reviewer 9tZD's insightful remarks on the specifics of our use case. The potential combination of adaptation and encoding cloning with a justification for its absence in our study and comments on fine-tuning techniques are now discussed as well.

To widen the evaluation scope, we conducted a perceptive test and added it to the paper.
Given the focus of this work on speaker similarity, we chose to perform a DMOS speaker similarity test, keeping only the automatic measure to assess naturalness.

Overall, the DMOS evaluation (discussed in section 5.1.2) confirms the differentiation between models trained on the closest and farthest speakers and ranks models trained on all speakers about the same as those trained on the closest voices (which were trained with almost 3x less data).

As reviewer 9tZD suggested, it is possible that the good performance of the model trained on all data in tests that focused on our atypical target speaker can be attributed to the additional data helping the model to learn a better representation of the speaker space, which proves useful in contexts where the target speaker is unusual. This is now clearly discussed in the paper.

Once again, we thank the reviewers for their useful comments and hope we were able to address their remarks as well as possible in the final version of the paper.

---

### Decision · Program_Chairs · 2023-06-14

**Decision:**

Accept

**Comment:**

SSW2003 received 45 papers. The acceptance rate is 82%. We are pleased to inform you that your paper has been accepted by the SSW2023 Program Committee. Please read the reviews carefully and submit your camera-ready paper by June 28th. Most of reviewers performed a detailed review. Please answer to their questions and take into account their comments.
Since your paper received a score below 5/9 that is strongly argued by the reviewers, note that the Program Committee will check if your manuscript has been significantly changed to specifically consider their remarks. Note that camera-ready papers are credited of one extra page to allow authors to consider reviewers’ suggestions. So max 7 pages in total including figures & refs.
The deadline for submitting the revised version (with full non anonymized authors and refs!) is 28th June.